# Physics-as-Inverse-Graphics: Unsupervised Physical Parameter Estimation from Video

**Miguel Jaques**
School of Informatics
University of Edinburgh
Edinburgh, UK
m.a.m.jaques@sms.ed.ac.uk

**Michael Burke**
School of Informatics
University of Edinburgh
Edinburgh, UK
michael.burke@ed.ac.uk

**Timothy Hospedales**
School of Informatics
University of Edinburgh
Edinburgh, UK
t.hospedales@ed.ac.uk

## Abstract

We propose a model that is able to perform unsupervised physical parameter estimation of systems from video, where the differential equations governing the scene dynamics are known, but labeled states or objects are not available. Existing physical scene understanding methods require either object state supervision, or do not integrate with differentiable physics to learn interpretable system parameters and states. We address this problem through a *physics-as-inverse-graphics* approach that brings together vision-as-inverse-graphics and differentiable physics engines, enabling objects and explicit state and velocity representations to be discovered. This framework allows us to perform long term extrapolative video prediction, as well as vision-based model-predictive control. Our approach significantly outperforms related unsupervised methods in long-term future frame prediction of systems with interacting objects (such as ball-spring or 3-body gravitational systems), due to its ability to build dynamics into the model as an inductive bias. We further show the value of this tight vision-physics integration by demonstrating data-efficient learning of vision-actuated model-based control for a pendulum system. We also show that the controller's interpretability provides unique capabilities in goal-driven control and physical reasoning for zero-data adaptation.

## 1 Introduction

System identification or physical parameter estimation is commonly required for control or state estimation for physical modelling, and typically relies on dedicated sensing equipment and carefully constructed experiments. Current machine learning approaches to physical modeling from video either require training by supervised regression from video to object coordinates before estimating explicit physics (Watters et al., 2017; Wu et al., 2017b; Belbute-Peres et al., 2018), or are able to discover and segment objects from video in an unsupervised manner, but do not naturally integrate with a physics engine for long-term predictions or generation of interpretable locations and physical parameters for physical reasoning (Xu et al., 2019; van Steenkiste et al., 2018). In this work, we bridge the gap between unsupervised discovery of objects from video and learning the physical dynamics of a system, by learning unknown physical parameters and explicit trajectory coordinates.

Our approach, called *physics-as-inverse-graphics*, solves the physical modeling problem via a novel vision-as-inverse-graphics encoder-decoder system that can render and de-render image components using Spatial Transformers (ST) (Jaderberg et al., 2015) in a way that makes it possible for the latent representation to generate disentangled interpretable states (position/velocity). These can be used

directly by a differentiable physics engine (Degrave et al., 2016; Belbute-Peres et al., 2018) to learn the parameters of a scene where the family of differential equations governing the system are known (e.g. objects connected by a spring), but the corresponding parameters are not (e.g. spring constant). This allows us to to identify physical parameters and learn vision components of the model jointly in an end-to-end fashion. Our contribution is a solution to unsupervised learning of physical parameters from video, without having access to ground-truth appearance, position or velocities of the objects, a task that had so far remained unsolved (Wu et al., 2015; Belbute-Peres et al., 2018).

In addition to showing that our model can learn physical parameters without object or state supervision (a task with intrinsic scientific interest in and of itself), we show that incorporating dynamics priors in the form of known physical equations of motion with learnable parameters together with learnable vision and graphics can improve model performance in two challenging tasks: long term video prediction and visual model predictive control. We first evaluate physical parameter estimation accuracy and future video frame prediction on 4 datasets with different non-linear interactions and visual difficulty. We then demonstrate the value of our method by applying it for data-efficient learning of vision-based control of an under-actuated pendulum. Notably our unique ability to extract interpretable states and parameters from pixels without supervision enables end-to-end vision-based control to exploit goal-paramaterized policies and physical reasoning for zero-shot adaptation.

## 2 RELATED WORK

The ability to build inductive bias into models through structure is a key factor behind the success of modern neural architectures. Convolutional operations capture spatial correlations (Fukushima, 1980) in images, recurrency allows for temporal reasoning (Hochreiter & Schmidhuber, 1997), and spatial transformers (Jaderberg et al., 2015) provide spatial invariance in learning. However, many aspects of common data generation processes are not yet considered by these simple inductive biases. Importantly, they typically ignore the physical interactions underpinning data generation. For example, it is often the case that the underlying physics of a dynamic visual scene is known, even if specific parameters and objects are not. Incorporation of this information would be beneficial for learning, predicting the future of the visual scene, or control. Physics-as-inverse graphics introduces a framework that allows such high-level physical interaction knowledge to be incorporated into learning, even when ground-truth object appearance, positions and velocities are not available.

In recent years there has been increased interest in physical scene understanding from video (Fragki-adaki et al., 2016; Finn et al., 2016; Fraccaro et al., 2017; Chang et al., 2017; Jonschkowski et al., 2017; Zheng et al., 2018; Janner et al., 2019). In order to learn explicit physical dynamics from video our system must discover and model the objects in a scene, having position as an explicit latent variable. Here we build on the long literature of neural vision-as-inverse-graphics (Hinton et al., 2011; Kulkarni et al., 2015; Huang & Murphy, 2016; Ellis et al., 2018; Romaszko et al., 2017; Wu et al., 2017a), particularly on the use of spatial transformers (ST) for rendering (Eslami et al., 2016; Rezende et al., 2016; Zhu et al., 2018).

There are several models that assume knowledge of the family of equations governing system dynamics, but where the individual objects are either pre-segmented or their ground-truth positions/velocities are known (Stewart & Ermon, 2017; Wu et al., 2017b; Belbute-Peres et al., 2018). In terms of learning physical parameters, our work is directly inspired by the Galileo model and Physics 101 dataset (Wu et al., 2015; 2016), which fits the dynamics equations to a scene with interacting objects. However, the Galileo model makes use of custom trackers which estimate the position and velocity of each object of interest, and is incapable of end-to-end learning from video, thus bypasses the difficulty of recognizing and tracking objects from video using a neural system. To the best of our knowledge, our model is the first to offer end-to-end unsupervised physical parameter and state estimation.

Within the differentiable physics literature (Degrave et al., 2016), Belbute-Peres et al. (2018) observed that a multi-layer perceptron (MLP) encoder-decoder architecture with a physics engine was not able to learn without supervising the physics engine's output with position/velocity labels (c.f. Fig. 4 in Belbute-Peres et al. (2018)). While in their case 2% labeled data is enough to allow learning, the transition to *no* labels causes the model to not learn at all. The key contribution of our work is the incorporation of vision-as-inverse-graphics with physics, which makes the transition possible.

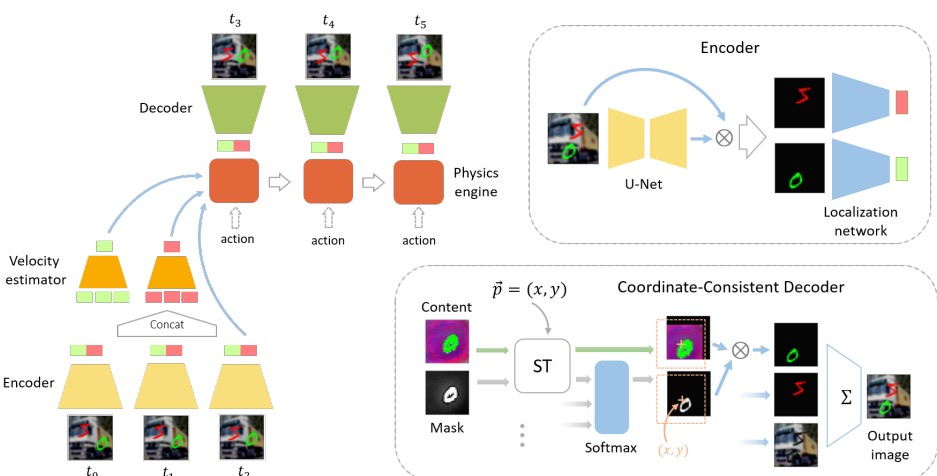

Figure 1: **Left:** High-level view of our architecture. The encoder (**top-right**) estimates the position of $N$ objects in each input frame. These are passed to the velocity estimator which estimates objects' velocities at the last input frame. The positions and velocities of the last input frame are passed as initial conditions to the physics engine. At every time-step, the physics engine outputs a set of positions, which are used by the decoder (**bottom-right**) to output a predicted image. If the system is actuated, an input action is passed to the physics engine at every time-step. See Section 3 for detailed descriptions of the encoder and decoder architectures.

Another related area of increasing interest is unsupervised discovery of objects and/or dynamics from video (Xu et al., 2019; van Steenkiste et al., 2018; Greff et al., 2019; Burgess et al., 2019). Though powerful, such models do not typically use interpretable latent representations that can be directly used by a physics engine, reasoned about for physical problem solving, or that are of explicit interest to model users. For example, Kosiorek et al. (2018) and Hsieh et al. (2018) use ST's to locate/place objects in a scene and predict their motion, but this work differs from ours in that our coordinate-consistent design obtains explicit cartesian, angular or scale coordinates, allowing us to feed state vectors directly into a differentiable physics engine. Under a similar motivation as our work, but without an inverse-graphics approach, Ehrhardt et al. (2018) developed an unsupervised model to obtain consistent object locations. However, this only applies to cartesian coordinates, not angles or scale.

Despite recent interest in model-free reinforcement learning, model-based control systems have repeatedly shown to be more robust and sample efficient (Deisenroth & Rasmussen, 2011; Mania et al., 2018; Watters et al., 2019a). Hafner et al. (2019) learn a latent dynamics model (PlaNet) that allows for planning from pixels, which is significantly more sample efficient than model-free learning strategies A3C (Mnih et al., 2016) and D4PG (Barth-Maron et al., 2018). However, when used for control, there is often a desire for visually grounded controllers operating under known dynamics, as these are verifiable and interpretable (Burke et al., 2019), and provide transferability and generality. However, system identification is challenging in vision-based control settings. Byravan et al. (2018) use supervised learning to segment objects, controlling these using known rigid body dynamics. Penkov & Ramamoorthy (2019) learn feedforward models with REINFORCE (Williams, 1992) to predict physical states used by a known controller and dynamical model, but this is extremely sample inefficient. In contrast, we learn parameter and state estimation modules jointly to perform unsupervised system identification from pixels, enabling data-efficient vision-actuated model-based control.

## 3 LEARNING PHYSICAL PARAMETERS FROM VIDEO VIA INVERSE GRAPHICS

In order to learn explicit physics from video, several components have to be in place. First, the model must be able to learn to identify and represent the objects in an image. In order to perform dynamics prediction with a physics engine, the position and velocity of the object must be represented

as explicit latent states (whereas appearance can be represented through some latent vector or, in our case, as a set of learned object templates). Our sequence-to-sequence video prediction architecture consists of 4 modules trained jointly: an encoder, a velocity estimator, a differentiable physics engine, and a graphics decoder. The architecture is shown in Figure 1.

**Encoder**  The encoder net takes a single frame $I_t$ as input and outputs a vector $\mathbf{p}_t \in \mathbb{R}^{N \times D}$ corresponding to the $D$-dimensional coordinates of each of $N$ objects in the scene, $\mathbf{p}_t = [\mathbf{p}_t^1, ..., \mathbf{p}_t^N]$. For example, when modelling position in 2D space we have $D = 2$ and $\mathbf{p}_t^n = [x, y]_t^n$; when modelling object angle we have $D = 1$ and $\mathbf{p}_t^n = [\theta_t^n]$. The encoder architecture is shown in Figure 1(top right).

To extract each object's coordinates we use a 2-stage localization approach[1]. First, the input frame is passed through a U-Net (Ronneberger et al., 2015) to produce $N$ unnormalized masks. These masks (plus a learnable background mask) are stacked and passed through a softmax to produce $N + 1$ masks, where each input pixel is softly assigned to a mask. The input image is then multiplied by each mask, and a 2-layer location network produces coordinate outputs from each masked input component. For a 2D system where the coordinates of each object are its $(x, y)$ position (the polar coordinates case is analogous) and the images have dimensions $H \times H$, the encoder output represents $(x, y)$ coordinates with values in $[0, H]$. To do this, the activation of the encoder's output layer is a saturating non-linearity $H/2 \cdot tanh(\cdot) + H/2$.

**Velocity estimator**  The velocity estimator computes the velocity vector of each object at the $L$-th input frame given the coordinates produced by the encoder for this object at the first $L$ input frames, $\mathbf{v}_L^n = f(\mathbf{p}_1^n, ..., \mathbf{p}_L^n)$. We implement this as a 3 hidden layer MLP with 100 tanh activated units.

**Differentiable physics engine**  The physics engine contains the differential equations governing the system, with unknown physical parameters to be learned – such as spring constants, gravity, mass, etc. Given initial positions and velocities produced by the encoder and velocity estimator, the physics engine rolls out the objects' trajectories. In this work we use a simple physics engine with Euler integration, where $\mathbf{p}_t, \mathbf{v}_t$ is computed from $\mathbf{p}_{t-1}, \mathbf{v}_{t-1}$ by repeating for $i \in [1..M]$:

$$\mathbf{p}_{t+\frac{i}{M}} = \mathbf{p}_{t+\frac{i-1}{M}} + \frac{\Delta t}{M} \cdot \mathbf{v}_{t+\frac{i}{M}} \quad ; \quad \mathbf{v}_{t+\frac{i}{M}} = \mathbf{v}_{t+\frac{i-1}{M}} + \frac{\Delta t}{M} \cdot \mathbf{F}(\mathbf{p}_{t+\frac{i-1}{M}}, \mathbf{v}_{t+\frac{i-1}{M}}; \theta), \quad (1)$$

where $\Delta t$ is the integration step, $\theta$ are the physical parameters and $\mathbf{F}$ is the force applied to each object, according to the equations in Appendix A. We use $M = 5$ in all experiments. In principle, more complex physics engines could be used (Chen et al., 2018; Belbute-Peres et al., 2018).

**Coordinate-Consistent Decoder**  The decoder takes as input the positions given by the encoder or physics engine, and outputs a predicted image $\tilde{I}_t$. The decoder is the most critical part of this system, and is what allows the encoder, velocity estimator and physics engine to train correctly in a fully unsupervised manner. We therefore describe its design and motivation in greater detail.

While an encoder with outputs in the range $[0, H]$ *can* represent coordinates in pixel space, it does not mean that the decoder *will* learn to correctly associate an input vector $(x, y)$ with an object located at pixel $(x, y)$. If the decoder is unconstrained, like a standard MLP, it can very easily learn erroneous, non-linear representations of this Cartesian space. For example, given two different inputs, $(x_1, y_1)$ and $(x_1, y_2)$, with $y_1 \neq y_2$, the decoder may render those two objects at different horizontal positions in the image. While having a correct Cartesian coordinate representation is not strictly necessary to allow physical parameters of the physics engine to be learned from video, it is critical to ensure correct future predictions. This is because the relationship between position vector and pixel space position must be fixed: if the position vector changes by $(\Delta x, \Delta y)$, the object's position in the output image must change by $(\Delta x, \Delta y)$. This is the key concept that allows us to improve on Belbute-Peres et al. (2018), in order to learn an encoder, decoder and physics engine without state labels.

In order to impose a correct latent-coordinate to pixel-coordinate correspondence, we use spatial transformers (ST) with inverse parameters as the decoder's writing attention mechanism. We want transformer parameters $\omega$ to be such that a decoder input of $\mathbf{p}_t^n = [x, y]_t^n$, places the center of the writing attention window at $(x, y)$ in the image, or that a decoder input of $\mathbf{p}_t^n = \theta_t^n$ rotates the attention window by $\theta$. In the original ST formulation (Jaderberg et al., 2015), the matrix $\omega$ represents the affine transformation applied to the *output* image to obtain the *source* image. This means that the elements of $\omega$ in Eq. 1 of Jaderberg et al. (2015) do not directly represent translation, scale or angle of

---

[1]Though any other architecture capable of effectively extracting object locations from images would work.

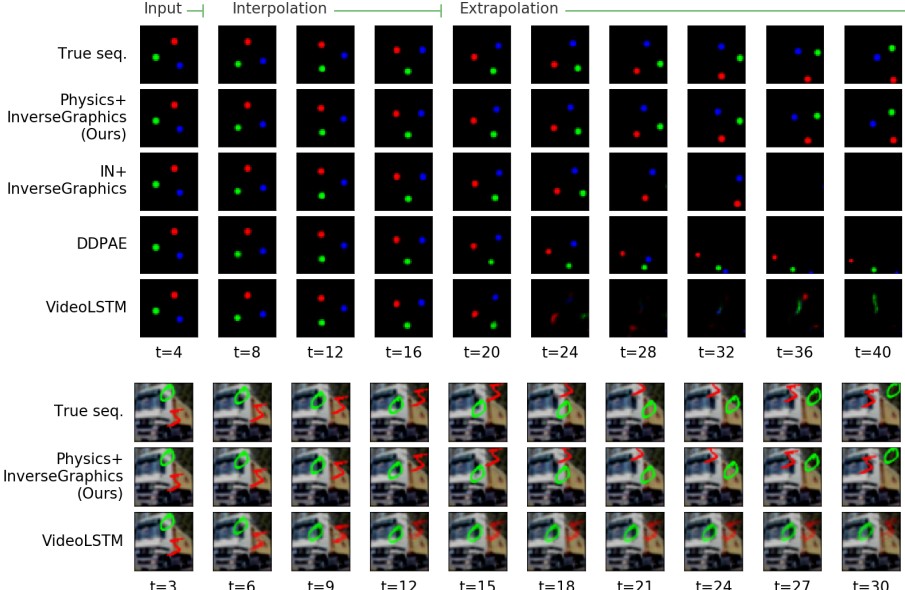

Figure 2: Future frame predictions for 3-ball gravitational system (**top**) and 2-digit spring system (**bottom**). IN: Interaction Network. Only the combination of Physics and Inverse-Graphics maintains object integrity and correct dynamics many steps into the future.

the writing attention window. To achieve this representation, we use a ST with inverse transformation parameters. For a general affine transformation with translation $(x, y)$, angle $\theta$ and scale $s$, we want to modify the source image coordinates according to:

$$\begin{pmatrix} x_o \\ y_o \\ 1 \end{pmatrix} = \begin{pmatrix} s \cdot \cos\theta & s \cdot \sin\theta & x \\ -s \cdot \sin\theta & s \cdot \cos\theta & y \\ 0 & 0 & 1 \end{pmatrix} \begin{pmatrix} x_s \\ y_s \\ 1 \end{pmatrix} \quad (2)$$

This transformation can be obtained with a ST by inverting (2):

$$\begin{pmatrix} x_s \\ y_s \\ 1 \end{pmatrix} = \frac{1}{s} \begin{pmatrix} \cos\theta & -\sin\theta & -x\cos\theta + y\sin\theta \\ \sin\theta & \cos\theta & -x\sin\theta - y\cos\theta \\ 0 & 0 & s \end{pmatrix} \begin{pmatrix} x_o \\ y_o \\ 1 \end{pmatrix} \quad (3)$$

Therefore, to obtain a decoder with coordinate-consistent outputs, we simply use a ST with parameters $\omega$ as given in (3)

Each object is represented by a learnable content $\mathbf{c}^n \in [0, 1]^{H \times H \times C}$ and mask $\mathbf{m}^n \in \mathbb{R}^{H \times H \times 1}$ tensor, $n = 1..N$. Additionally, we learn background content $\mathbf{c}^{bkg} \in [0, 1]^{H \times H \times C}$ and mask $\mathbf{m}^{bkg} \in \mathbb{R}^{H \times H \times 1}$, that do not undergo spatial transformation. One may think of the content as an RGB image containing the texture of an object and the mask as a grayscale image containing the shape and z-order of the object. In order to produce an output image, the content and mask are transformed according to $[\hat{\mathbf{c}}_t^n, \hat{\mathbf{m}}_t^n] = \mathrm{ST}([\mathbf{c}^n, \mathbf{m}^n], \omega_{\mathbf{P}_t^n})$ and the resulting logit masks are combined via a softmax across channels, $[\tilde{\mathbf{m}}_t^1, ..., \tilde{\mathbf{m}}_t^N, \tilde{\mathbf{m}}_t^{bkg}] = \mathrm{softmax}(\hat{\mathbf{m}}_t^1, ..., \hat{\mathbf{m}}_t^N, \mathbf{m}^{bkg})$. The output image is obtained by multiplying the output masks by the contents:

$$\tilde{I}_t = \tilde{\mathbf{m}}_t^{bkg} \odot \mathbf{c}^{bkg} + \sum_{n=1}^{N} \tilde{\mathbf{m}}_t^n \odot \hat{\mathbf{c}}_t^n. \quad (4)$$

The decoder architecture is shown in Fig. 1, bottom-right. The combined use of ST's and masks provides a natural way to model depth ordering, allowing us to capture occlusions between objects.

**Auxiliary autoencoder loss** Using a constrained decoder ensures the encoder and decoder produces objects in consistent locations. However, it is hard to learn the full model from future frame prediction alone, since the encoder's training signal comes exclusively from the physics engine. To alleviate this and quickly build a good encoder/decoder representation, we add a static per-frame autoencoder loss.

**Training** During training we use $L$ input frames and predict the next $T_{pred}$ frames. Defining the frames produced by the decoder via the physics engine as $\tilde{I}_t^{\text{pred}}$ and the frames produced by the decoder using the output of the encoder directly as $\tilde{I}_t^{\text{ae}}$, the total loss is:

$$\mathcal{L}_{total} = \mathcal{L}_{\text{pred}} + \alpha\mathcal{L}_{\text{rec}} = \sum_{t=L+1}^{L+T_{pred}} \mathcal{L}(\tilde{I}_t^{\text{pred}}, I_t) + \alpha \sum_{t=1}^{L+T_{pred}} \mathcal{L}(\tilde{I}_t^{\text{ae}}, I_t) \qquad (5)$$

where $\alpha$ is a hyper-parameter. We use mean-squared error loss throughout. During testing we predict an additional $T_{ext}$ frames in order to evaluate long term prediction beyond the length seen for training.

## 4 EXPERIMENTS

### 4.1 PHYSICAL PARAMETER LEARNING AND FUTURE PREDICTION

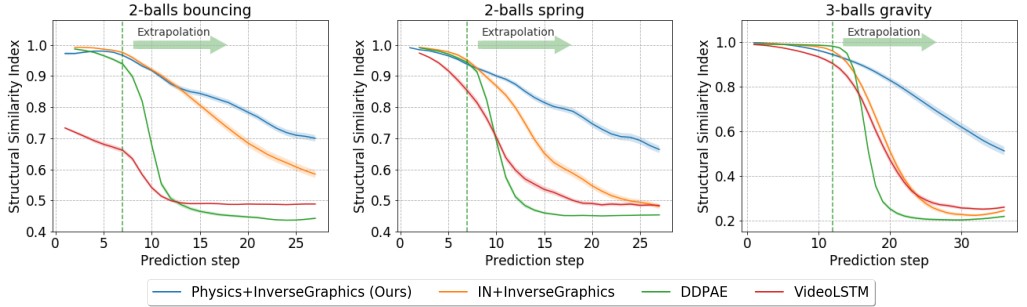

Figure 3: Frame prediction accuracy (SSI, higher is better) for the balls datasets. Left of the green dashed line corresponds to the training range, $T_{pred}$, right corresponds to extrapolation, $T_{ext}$. We outperform Interaction Networks (IN) (Watters et al., 2017), DDPAE (Hsieh et al., 2018) and VideoLSTM (Srivastava et al., 2015) in extrapolation due to incorporating explicit physics.

| Dataset | 2-balls spring | 2-digits spring | 3-balls gravity | 3-balls gravity |
|---|---|---|---|---|
| Parameters | $(k, l)$ | $(k, l)$ | $g$ | $m$ |
| Learned value | (4.26, 6.17) | (2.18, 12.24) | 65.7 | 0.95 |
| Ground-truth value | (4.0, 6.0) | (2.0, 12.0) | 60.0 | 1.0 |

Table 1: Physical parameters learned from video are within 10% of true system parameters.

**Setup** To explore learning physical parameters and evaluate long-term prediction we train our model on scenes with 5 different settings: two colored balls bouncing off the image edges; two colored balls connected by a spring; three colored balls with gravitational pull – all on a black background; and to test greater visual complexity, we also use 2 MNSIT digits connected by a spring, on a CIFAR background. We train using values of $(L, T_{pred}, T_{ext})$ set to $(3, 7, 20)$, $(3, 7, 20)$, $(3, 7, 20)$, $(4, 12, 24)$ and $(3, 7, 20)$, respectively. For the spring systems the physical parameters to be learned are the spring constant $k$ and equilibrium distance $l$, and for the gravitational system it is the gravity constant $g$ or mass of the objects $m$ (when learning gravity the mass if fixed, and vice-versa). In all cases we use objects with mass $m = 1$. We provide the exact equations of motion used in these systems and other training details in Appendices A and B, respectively. All datasets consist of 5000 sequences for training, 500 for validation, and 500 for testing. We use a learnable ST scale parameter initialized at $s = 2$ in the balls datasets and $s = 1$ in the digits dataset. In these datasets we set $\theta = 0$.

**Baselines** We compare our model to 3 strong baselines: DDPAE (Hsieh et al., 2018)[2], which is a generative model that uses an inverse-graphics model with black-box dynamics; VideoLSTM (Srivastava et al., 2015), which uses black-box encoding, decoding and dynamics; Interaction Network + Inverse-Graphics, which uses the same encoder and decoder as our Physics-as-Inverse-Graphics model, but where the dynamics module is an Interaction Network (Battaglia et al., 2016). The latter

---

[2]Using the code provided by the authors.

model allows us to compare explicit physics with relational dynamics networks, in terms of their ability to correctly capture object interactions[3].

**Results**  Table 1 shows that our model finds physical parameters close to the ground-truth values used to generate the datasets, and Figure 4 shows the contents and masks learned by the decoder. This highlights the fact that the proposed model can successfully perform unsupervised system identification from pixels. Future frame predictions for two of the systems are shown in Figure 2, and per-step Structural Similarity Index (SSI) [4] of the models on the prediction and extrapolation range are shown in Figure 3. While all models obtain low error in the prediction range (left of the green dashed line), our model is significantly better in the extrapolation range. Even many steps into the future, our model's predictions are still highly accurate, unlike those of other black-box models (Figure 2). This shows the value of using an explicit physics model in systems where the dynamics are non-linear yet well defined. Further rollouts are shown in Appendix C, and we encourage the reader to watch the videos for all the datasets at `https://sites.google.com/view/physicsasinversegraphics`.

This difference in performance is explained in part by the fact that in some of these systems the harder-to-predict parts of the dynamics do not appear during training. For example, in the gravitational system, whiplash from objects coming in close contact is seldom present in the first $K + T_{pred}$ steps given in the training set, but it happens frequently in the $T_{ext}$ extrapolation steps evaluated during testing. We do not consider this to be a failure of black-box model, but rather a consequence of the generality *vs* specificity tradeoff: a model without a sufficiently strong inductive bias on the dynamics is simply not able to correctly infer close distance dynamics from long distance dynamics.

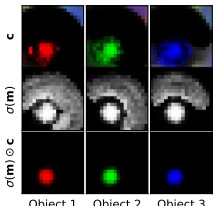
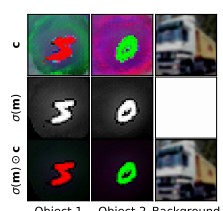

| Train using | $\mathcal{L}_{\text{pred}}$ | $\mathcal{L}_{\text{rec}}$ |
|---|---|---|
| only $\mathcal{L}_{\text{pred}}$ | 31.4 | 20.5 |
| separate gradients | 28.1 | 0.22 |
| joint $\mathcal{L}_{\text{pred}} + \alpha\mathcal{L}_{\text{rec}}$ | **1.39** | 0.63 |
| black-box decoder, joint | 30.9 | 2.87 |

Figure 4: Contents and masks learned by the decoder. Object masks: $\sigma(\mathbf{m})$. Objects for rendering: $\sigma(\mathbf{m}) \odot \mathbf{c}$. Contents and masks correctly capture each part of the scene: colored balls, MNIST digits and CIFAR background. We omit the black background learned on the balls dataset.

Table 2: Test loss under different training conditions. Separate gradients: Train encoder/decoder on $\mathcal{L}_{\text{rec}}$, and velocity estimator and physics engine on $\mathcal{L}_{\text{pred}}$. Black-box decoder, joint: Joint training using a standard MLP network as the decoder. Only joint training using our coordinate-consistent decoder succeeds.

**Ablation studies**  Since the encoder and decoder must discover the objects present in the image and the corresponding locations, one might assume that the velocity estimator and physics engine could be learned using only the prediction loss, and encoder/decoder using only the static autoencoder loss, i.e., without joint training. In Table 2 we compare the performance of four variants on the 3-ball gravity dataset: joint training using only the prediction loss; joint training using the prediction and autoencoder losses; training the encoder/decoder on the autoencoder loss and the velocity estimator and physics engine on the prediction loss; and joint training, but using an MLP black-box decoder.

We can see that only joint prediction and autoencoder loss obtain satisfactory performance, and that the use of the proposed coordinate-consistent decoder is critical. The prediction loss is essential in order for the model to learn encoders/decoders whose content and masks can be correctly used by the physics engine. This can be understood by considering how object interaction influences the decoder. In the gravitational system, the forces between objects depend on their distances, so if the objects swap locations, the forces must be the same. If the content/mask learned for each object are centered differently relative to its template center, rendering the objects at positions $[x, y]$ and $[w, z]$, or $[w, z]$ and $[x, y]$ will produce different distances between these two objects in image space. This violates the

---

[3]This baseline also serves as strong proxy for comparison with recent relational models (Watters et al., 2017; van Steenkiste et al., 2018), which due to their supervision method or input-output space cannot be directly compared our model.

[4]We choose SSI over MSE as an evaluation metric as it is more robust to pixel-level differences and alignment.

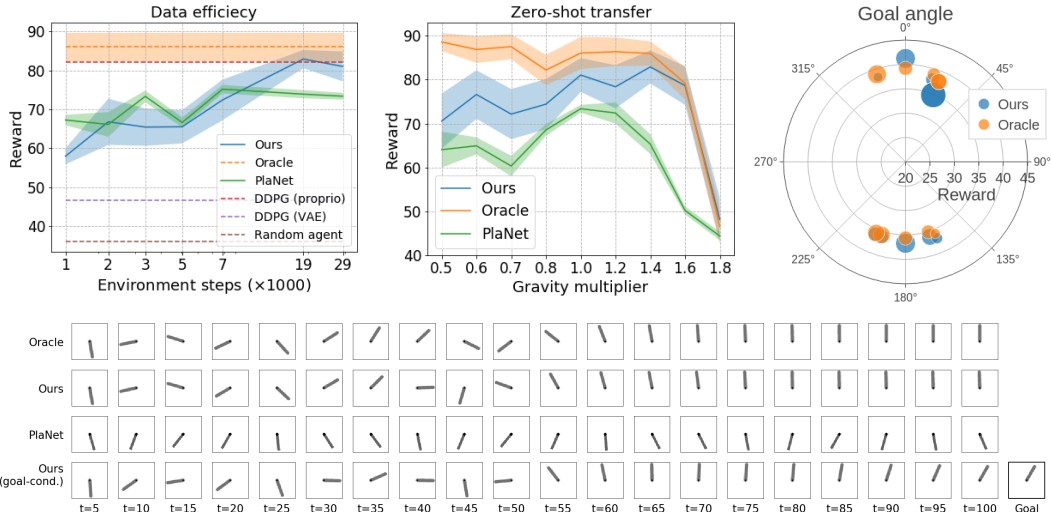

Figure 5: **Top:** Comparison between our model and PlaNet Hafner et al. (2019) in terms of learning sample efficiency **(left)**. Explicit physics allows reasoning for zero-shot adaptation to domain-shift in gravity **(center)** and goal-driven control to balance the pendulum in any position **(right)**. DDPG (VAE) corresponds to a DDPG agent trained on the latent space of an autoencoder (trained with 320k images) after 80k steps. DDPG (proprio) corresponds to an agent trained from proprioception after 30k steps. **Bottom:** The first 3 rows show a zero-shot counterfactual episode with a gravity multiplier of 1.4 for an oracle, our model and planet, with vertical as the target position (as trained). The last row shows an episode using a goal image to infer the non-vertical goal state.

permutation invariance property of the system. Learning the encoder/decoder along with the velocity estimator and physics engine on the prediction loss allows the encoder and decoder to learn locations and contents/masks that satisfy the characteristics of the system and allows the physics to be learned correctly. In Appendix D we perform further ablations on the decoder architecture and its ability to correctly render objects in regions of the image not seen during training.

## 4.2 VISION-BASED MODEL-PREDICTIVE CONTROL (MPC)

**Tasks** One of the main applications of our method is to identify the (actuated) dynamical parameters and states of a physical system from video, which enables vision-based planning and control. Here we apply it to the pendulum from OpenAI Gym (Brockman et al., 2016) – one typically solved from proprioceptive state, not pixels. For training we collect 5000 sequences of 14 frames with random initialization ($\dot{\theta}_0 \sim \text{Unif}(-6, 6)$) and actions ($u_t \sim \text{Unif}(-2, 2)$). The physical parameters to learn are gravity $g = 10.0$ and actuation coefficient $a = 1.0$. We use $K = 4$ and $T_{pred} = 10$. We use the trained MPC model as follows. At every step, the previous 4 frames are passed to the encoder and velocity nets to estimate $[\theta_t, \dot{\theta}_t]$. This is passed to the physics engine with learned parameters $g$ and $a$. We perform 100-step model-predictive control using the cross entropy method (Rubinstein, 1997), exactly as described in Hafner et al. (2019), setting vertical position and zero velocity as the goal. **Baselines** We compare our model to an oracle model, which has the true physical parameters and access to the true pendulum position and velocity (not vision-based), as well as a concurrent state-of-the art model-based RL method (PlaNet (Hafner et al., 2019)), and a model-free[5] deep deterministic policy gradient (DDPG) agent (Lillicrap et al., 2016). To provide an equivalent comparison to our model, we train PlaNet on random episodes.

**Results** In terms of system identification, our model recovers the correct gravity ($g = 9.95$) and force coefficient ($a = 0.99$) values from vision alone, which is a prerequisite to perform correct planning and control. Figure 5 (top-left) highlights the data efficiency of our method, which is comparable to PlaNet, while being dramatically faster than DDPG from pixels. Importantly, the

---

[5]DDPG, TRPO and PPO learned from pixels failed to solve the pendulum, highlighting the complexity of the vision-based pendulum control task and brittleness of model-free reinforcement learning strategies.

interpretability of the explicit physics in our model provides some unique capabilities. We can perform simple counter-factual physical reasoning such as '*How should I adapt my control policy if gravity was increased?*', which enables zero-shot adaptation to new environmental parameters. Figure 5 (top-middle) shows that our model can exploit such reasoning to succeed immediately over a wide range of gravities with no re-training. Similarly, while the typical inverted pendulum goal is to balance the pendulum upright, interpretable physics means that this is only one point in a space of potential goals. Figure 5 (top-right) evaluates the goal-paramaterized control enabled by our model. Any feasible target angle specified can be directly reached by the controller. There is extrapolative generalisation across the space of goals even though only one goal (vertical) was seen during training. Importantly, these last two capabilities are provided automatically by our model due to its disentangled interpretable representation, but cannot be achieved without further adaptive learning by alternatives that are reward-based (Mnih et al., 2016) or rely on implicit physics (Hafner et al., 2019).

## 5 LIMITATIONS

Alhough the approach presented here shows promising results in terms of physical parameter estimation, long-term video prediction and MPC, a number of limitations need to be overcome for real-world application.

**Templates as object representation** Though the assumption that every scene in a dataset is a combination of learnable templates is a common one in the literature (c.f. Tieleman (2014) for an extensive study on this), this is insufficient to model real-world scenes. For example, applying physics-as-inverse-graphics to the Physics101 dataset would require representing objects using a latent appearance representation that could be used by the decoder (Eslami et al., 2016). This would introduce new modelling challenges, requiring object tracking to keep correct object identity associations (Kosiorek et al., 2018). In this work we simplify this problem by assuming that objects are visually distinct throughout the dataset, though this does not detract from the essential contributions of the paper.

**Rigid sequence to sequence architecture** In this work we used a sequence-to-sequence architecture, with a fixed number of input steps. This architectural choice (inspired by Watters et al. (2017)), prevents the model from updating its state beliefs if given additional input frames later in the sequence. Formulating the current model in a probabilistic manner that would allow for state/parameter filtering and smoothing at inference time is a promising direction of future work.

**Static background assumption** Many scenes of interest do not follow the assumption that the only moving objects in the scene are the objects of interest (even though this assumption is widely used). Adapting our model to varying scene backgrounds would require additional components to discern which parts of the scene follow the dynamics assumed by the physics engine, in order to correctly perform object discovery. This is a challenging problem, but we believe it would greatly increase the range of applications of the ideas presented here.

## 6 CONCLUSION

Physics-as-inverse graphics provides a valuable mechanism to integrate inductive bias about physical data generating processes into learning. This allows unsupervised object tracking and system identification, in addition to sample efficient, generalisable and flexible control. However, incorporating this structure into lightly supervised deep learning models has proven challenging to date. We introduced a model that accomplishes this, relying on a coordinate-consistent decoder that enables image reconstruction from physics. We have shown that our model is able to perform accurate long term prediction and that it can be used to learn the dynamics of an actuated system, allowing us to perform vision-based model-predictive control.

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

## A    SYSTEM DESCRIPTIONS

In this section we describe the equations of motion used for each system.

**2-balls and 2-digits spring**  The force applied on object $i$ by object $j$ follows Hooke's law:

$$\vec{F}_{i,j} = -k\,(\vec{p}_i - \vec{p}_j) - l\frac{\vec{p}_i - \vec{p}_j}{|\vec{p}_i - \vec{p}_j|}.$$

(6)

Each step corresponds to an interval $\Delta t = 0.3$.

**3-balls gravity**  The force applied on object $i$ by object $j$ follows Newton's law of gravity:

$$\vec{F}_{i,j} = -g\,m_i m_j \frac{\vec{p}_i - \vec{p}_j}{|\vec{p}_i - \vec{p}_j|^3}$$

(7)

where the masses are set to 1. Each step corresponds to an interval $\Delta t = 0.5$.

**Pendulum**  The pendulum follows the equations used by the OpenAI Gym environment:

$$\vec{F} = -\frac{3}{2}g\sin(\theta + \pi) + 3u$$

(8)

where $u$ is the action. Each step corresponds to an interval $\Delta t = 0.05$. In the physics engine used by the model we introduce an extra actuation coefficient $a$ to be learned along with $g$:

$$\vec{F} = -\frac{3}{2}g\sin(\theta + \pi) + a \cdot u$$

(9)

## B    TRAINING DETAILS AND HYPERPARAMETERS

For all datasets we use RMSProp Hinton et al. (2012) with an initial learning rate of $3 \times 10^{-4}$. For the balls and digits datasets we train for 500 epochs with $\alpha = 2$, and divide the learning rate by 5 after 375 epochs. For the pendulum data we train for 1000 epochs using $\alpha = 3$, but divide the learning rate by 5 after 500 epochs. The image sizes are $32 \times 32$ for the 2-balls bouncing and spring, $36 \times 36$ for the 3-balls gravity, $64 \times 64$ for the 2-digits spring, and $64 \times 64$ grayscale for the pendulum.

The content and mask variables are the output of a neural network with a constant array of 1s as input and 1 hidden layer with 200 units and tanh activation. We found this easier to train rather than having the contents and masks as trainable variables themselves.

## C  ADDITIONAL ROLLOUTS FOR EACH DATASET

### 3-BALLS GRAVITY

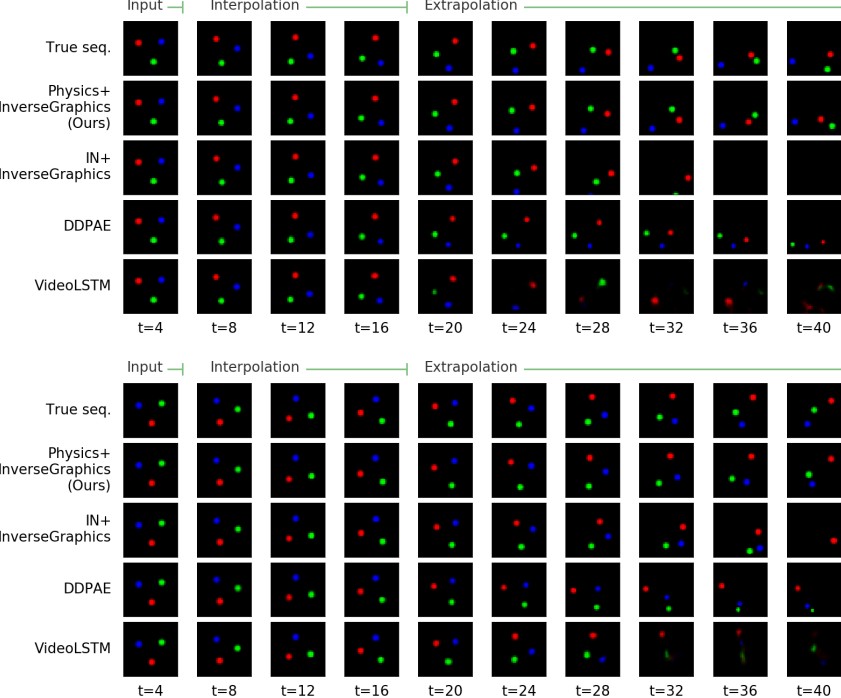

### 2-BALLS SPRING

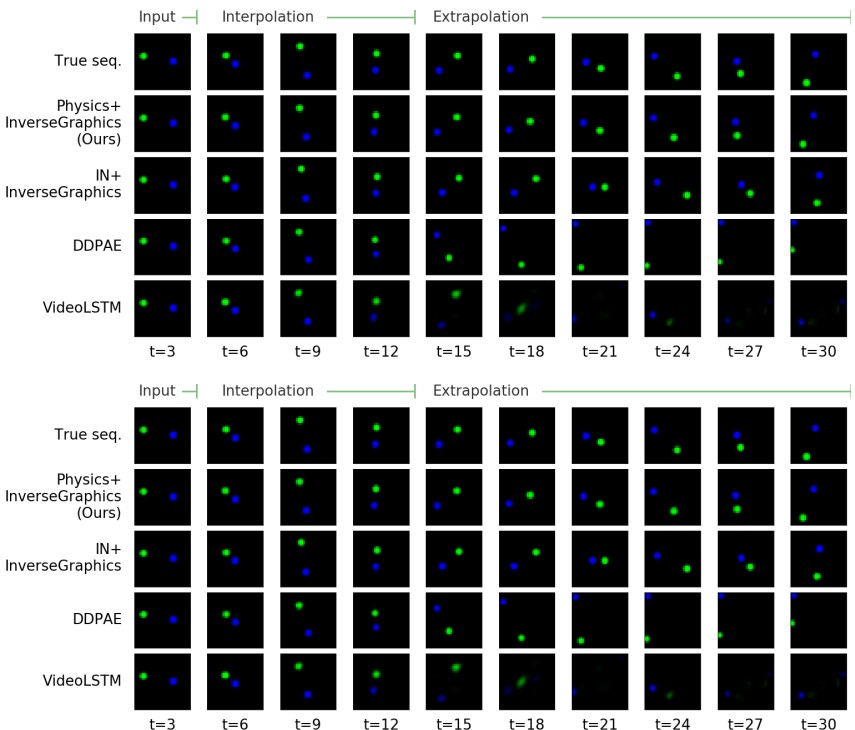

2-BALLS BOUNCING

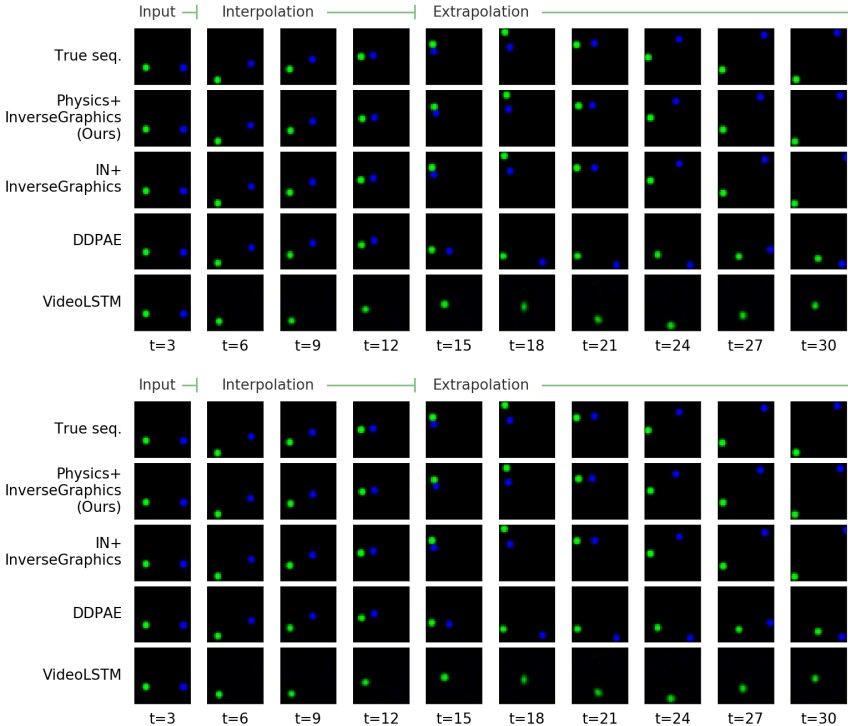

2-DIGITS SPRING

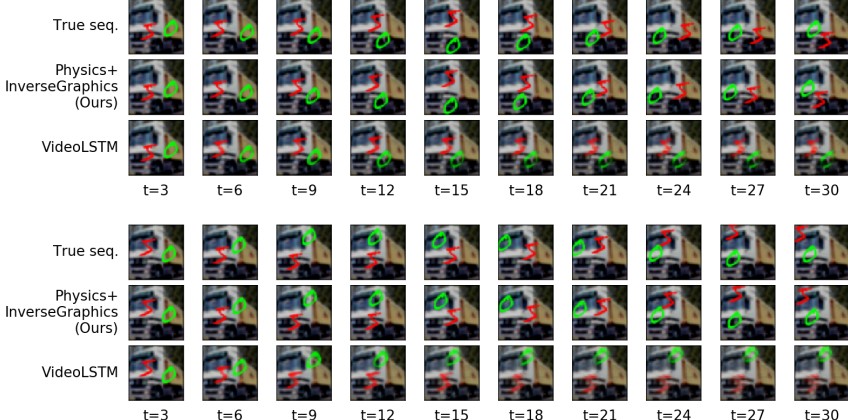

## D    EXTRAPOLATION TO UNSEEN IMAGE REGIONS

One limitation of standard fully-connected or deconvolutional decoders is the inability to decode states corresponding to object poses or locations not seen during training. For example, if in the training set no objects appear in the bottom half of the image, a fully-connected decoder will simply learn to output zeros in that region. If in the test set objects move into the bottom half of the image, the decoder lacks the inductive bias necessary to correctly extrapolate in image space.

To test this hypothesis, we replaced our model's decoder with a Deconv and Spatial Broadcast (Watters et al., 2019b) decoder, and compared them in a spatial extrapolation experiment. In this experiments, objects never enter the bottom half of the image in the input and prediction range, though in the extrapolation range in the test set objects move to this region of the scene. In the

rollouts shown in Fig. D, Broadcast performs better than Deconv, but they both fail to maintain object integrity when the balls move to the bottom half of the image in the extrapolation steps, validating our hypothesis that a black-box decoder has insufficient extrapolation ability. In contrast, our rendering decoder is be able to correctly decode states not seen during training.

In the limit that our renderer corresponds to a full-blown graphics-engine, any pose, location, color, etc. not seen during training can still be rendered correctly. This property gives models using rendering decoders, such as ours and Hsieh et al. (2018), an important advantage in terms of data-efficiency. We note, however, that in general this advantage does not apply to correctly inferring the states from images whose objects are located in regions not seen during training. This is because the encoders used are typically composed simply of convolutional and fully-connected layers, with limited de-rendering inductive biases.

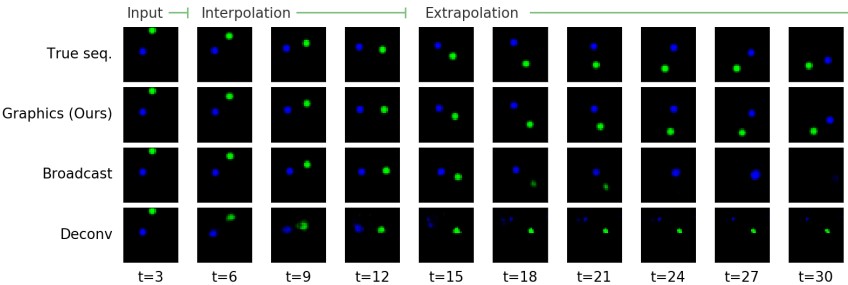

Figure 6: Comparison between graphics decoder and two black-box decoders, trained on data where objects only appear in the top half of the scene. Only the graphics decoder is able to correctly render the objects in the bottom half of the scene at test time. Broadcast: spatial broadcast decoder (Watters et al., 2019b); Deconv: standard deconvolutional network.

## E    INCORRECT NUMBER OF OBJECT SLOTS

The model proposed assumes we know the number of objects present in the scene. Here we briefly explore how to the model behaves when we use an incorrect number of slots $N$. We use the gravitational system, since interaction forces between objects are easy to generalize for any $N$. Fig. 7, left, shows that when using only 2 object slots, two of the objects are found, since the model does not have capacity to find more. Fig. 7, right, shows that when using more slots than the number of objects in the scene, all objects are discovered, and extra slots are left empty. However, in both cases we found predictive performance to be subpar, since in one case there are objects missing to correctly infer interactions, and in the other there are interactions between object slots and empty slots, confusing the dynamics.

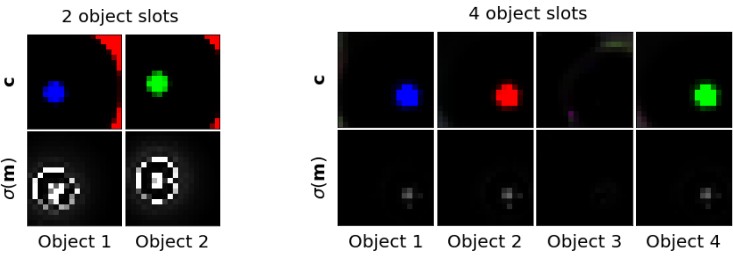

Figure 7: Results for incorrect number of object slots in the physics engien for the 3-body gravitational system **Left:** Contents and masks learned for 2 object slots. **Right:** Contents and objects learned for 4 object slots.

