# OpenReview forum: "Physics-as-Inverse-Graphics: Unsupervised Physical Parameter Estimation from Video"
_ICLR.cc/2020/Conference — Accept (Poster)_

### Official Review · AnonReviewer3 · 2019-10-16
**Official Blind Review #3**

**Rating:** 6

**Review:**

The paper proposes to integrate model-based physical simulation and data-driven (deep) learning. In a nutshell, one deep network predicts the state variables of the physics simulation (such as objet location, shape and velocity) from an image. A second network does the inverse task, to render images given the state variables (and a background image). In this way, one can go from a video frame to a physical system state, modify the state with physics simulation, and then go back from the modified state to a video frame. Together with a differentiable physics engine, through which one can back-propagate, this makes it possible to use the un-annotated video itself as supervision. At the same time, the two neural networks can be seen as an auto-encoder, in which the latent state is explicitly constrained to correspond to the desired physical state variables.

The topic of the paper is hot: a proper integration of physical models with data-driven deep learning is, arguably, one of the big short- to mid-term themes of machine learning research. The way it is done in the present paper intuitively makes sense. The approach is fairly obvious at the conceptual level; but in the details poses a number of technical challenges especially for the decoder, which are nicely analysed and resolved.

Some minor design choices are not well justified and at first sight appear a bit l'art pour l'art. While it is a sensible, pragmatic choice to first predict object masks, then extract their location ands and velocities in a second step; I do not quite see why one would have to do the latter with neural networks. it would seem that once the masks have been found, their location can be chosen as something like the (perfectly differentiable) mask-weighted centroid and does not need a multi-layer network; and similarly that deriving velocity from locations in adjacent frames can be hard-coded and does not need a 3-layer network.

The experiments are still at an early "toy" level, with synthetic videos where high-contrast, homogeneous objects move in front of a uniform or blurry background. The baselines are sensible and ablation studies are done with care. Still, it would have been nice to also run the method on some real video. To my understanding, this would be easily possible at least for future frame prediction, all one has to do is either annotate the objects in the target frame or measure success by comparing the predicted and true frames at the image level. It is also not clear whether the videos were synthesised with the same physics engine also use inside the system - which would be slightly questionable, in the sense that the learnable pipeline is then a-priori matched to the biases in the data.

One comment on the presentation: while the paper is generally well-written and easy to follow, the wording could at times be more careful. There is a slight tendency to identify the particular (simple) physical systems of the paper with physics as a whole. E.g., not all physics simulation must have objects - for instance, fluid dynamics or radiative transfer do not have individual objects, but are nevertheless relevant in the context of visual data. Similarly, even for defined objects, position and velocity are not always a sufficient state, for instance objects might deform, or have different elastic properties when colliding.

Overall, I find the work interesting and well-executed. It is a natural step to take towards the important goal of integrated data-driven and physical models, including the associated theme of self-supervision via physical constraints. On the negative side the paper does make a slightly rushed and unfinished impression by not showing any, even qualitative, experiments on real video. Most people - rightly - use simple toy-like datasets for development and analysis. But showing only those gives me the impression that the paper was written too early, just to be the first and to make the deadline. Or that moving to real video poses a much greater challenge than expected - but then this should be stated and discussed.


**Experience Assessment:**

I have read many papers in this area.

**Review Assessment: Checking Correctness Of Derivations And Theory:**

I assessed the sensibility of the derivations and theory.

**Review Assessment: Checking Correctness Of Experiments:**

I assessed the sensibility of the experiments.

**Review Assessment: Thoroughness In Paper Reading:**

I read the paper at least twice and used my best judgement in assessing the paper.

---

> ### Author Response · Authors · 2019-11-13
> **Addressing Reviewer 3' questions**
>
> Q1: Some minor design choices are not well justified and at first sight appear a bit l'art pour l'art. While it is a sensible, pragmatic choice to first predict object masks, then extract their location ands and velocities in a second step; I do not quite see why one would have to do the latter with neural networks. it would seem that once the masks have been found, their location can be chosen as something like the (perfectly differentiable) mask-weighted centroid and does not need a multi-layer network; and similarly that deriving velocity from locations in adjacent frames can be hard-coded and does not need a 3-layer network.
>
> Q1: We agree that it might be possible to generate coordinate consistent object positions in alternative ways. We used an encoder that outputs the affine parameters for a spatial transformer since it provides an elegant way to combine shift, scale and rotation transformation.
> Regarding initial velocity computation, during experiments we noticed that using an MLP, though not an elegant solution, provided better initial velocity estimates particularly in the gravitational system and pendulum cases. We believe this is because the initial velocity in this case depends on the acceleration (and higher moments) of the objects, which an MLP can more easily capture, given object positions.
>
> Q2: The experiments are still at an early "toy" level. , with synthetic videos where high-contrast, homogeneous objects move in front of a uniform or blurry background. The baselines are sensible and ablation studies are done with care. Still, it would have been nice to also run the method on some real video. To my understanding, this would be easily possible at least for future frame prediction, all one has to do is either annotate the objects in the target frame or measure success by comparing the predicted and true frames at the image level. It is also not clear whether the videos were synthesised with the same physics engine also use inside the system - which would be slightly questionable, in the sense that the learnable pipeline is then a-priori matched to the biases in the data.
>
> A2: We address this in the main comment.
>
> Q3: One comment on the presentation: while the paper is generally well-written and easy to follow, the wording could at times be more careful. There is a slight tendency to identify the particular (simple) physical systems of the paper with physics as a whole. E.g., not all physics simulation must have objects - for instance, fluid dynamics or radiative transfer do not have individual objects, but are nevertheless relevant in the context of visual data. Similarly, even for defined objects, position and velocity are not always a sufficient state, for instance objects might deform, or have different elastic properties when colliding.
>
> A3: This is a good point. We will try to change the wording in order to avoid sweeping statements about physical simulation as a research area.
>
> (continues)

---

> ### Author Response · Authors · 2019-11-13
> **Addressing Reviewer 3's questions (cont.)**
>
> Q4: Overall, I find the work interesting and well-executed. It is a natural step to take towards the important goal of integrated data-driven and physical models, including the associated theme of self-supervision via physical constraints. On the negative side the paper does make a slightly rushed and unfinished impression by not showing any, even qualitative, experiments on real video. Most people - rightly - use simple toy-like datasets for development and analysis. But showing only those gives me the impression that the paper was written too early, just to be the first and to make the deadline. Or that moving to real video poses a much greater challenge than expected - but then this should be stated and discussed.
>
> A4: We agree that more space could have been allocated to discussing the limitations of the proposed approach and have included a section discussing these in greater detail. Although it may seem that physical equations provides a strong inductive bias that simplifies learning, this is definitely not the case, and learning object representations, states and physical equation parameters that satisfy the constraints of physical equations is particularly challenging, and required very careful architectural and representation choices for us to achieve this. Indeed, learning with constraints is much harder than learning without these constraints, but, as our results show, provides valuable performance gains when successful. We believe that the complexity of learning with physical constraints is the primary reason that no existing unsupervised visual system identification approaches have been developed thus far. However, we do acknowledge that there is substantial room for improvement if the proposed approach is to move beyond toy examples and towards general, real world video. Importantly, this requires substantial improvements in model rendering and derendering components. We discuss these challenges in more detail in the revised paper, but believe that the model proposed in this work is an important waypoint if this goal is to be realised.

---

### Official Review · AnonReviewer1 · 2019-10-23
**Official Blind Review #1**

**Rating:** 6

**Review:**

This paper presents an approach for unsupervised estimation of physical parameters from video, using the physics as inverse graphics approach. The approach uses a feedforward encoder for localising object positions from which a velocity is estimated. These are fed as inputs to an (action-conditioned) physics simulator which generates future predictions of object positions. This simulator has knowledge of the system dynamics apriori, needing estimation of a few physical parameters such as gravity and spring constants. The outputs of this simulator is fed to a co-ordinate consistent decoder, a neural network that uses a Spatial Transformer to render the corresponding output image. The whole system is trained end-to-end on videos of dynamical systems, in an unsupervised manner. Results on two and three body interaction settings and an MNIST digit motion dataset show promising performance. The system is able to recover the underlying physical parameters accurately while also making consistent long-term ex predictions. Additionally, the model is used for visual MPC on a simulated cartpole task where it outperforms state of the art model-based and model-free RL baselines.

This paper is well written and clearly motivated, albeit a bit incremental in its approach. Many of the building blocks have been explored in prior work, with the major component being the co-ordinate consistent decoder. The experiments are visually simplistic and it is not obvious if the system will scale to more complex settings. A few comments:
1. A major limitation of the approach is the assumption that the equations governing the system are known. This makes it harder to generalise the system to novel tasks and more complicated settings with contact and object interactions. A potential way to overcome this could be to use ideas from prior work such as Interaction Networks where the dynamics are modelled as unary and binary interactions. While such dynamics models are black-box and not as interpretable compared to the current approach, they can easily generalise to novel tasks. Additionally, using positions and velocities as the latent state representation together with IN style transition models can be a sensible middle ground.
2. It would be great if there is an additional ablation experiment where the known equations of motion are replaced with a black-box neural network (while still retaining the position and velocity representation). This can quantify the effect of known dynamics and make the contributions of the paper (with regards to the decoder) more clear.
3. An alternative way of generating consistent object positions from the encoder is to compute a mask-weighted average of the image co-ordinates. This can be a nice way of adding additional structure to the networks that can regularise training.
4. The approach uses a 3-layer MLP for generating velocity estimates — could this not be done via finite differencing? (e.g. higher-order backward differencing)
5. Both the content and mask vectors are learnable but fixed for the entire task — i.e. it is not a function of the input image. This makes the approach not applicable to novel objects or even objects with minor color changes.
6. It is not clear how the translations, rotations and scale parameters for the Spatial Transformer are estimated. I presume that the positions and orientations predicted by either the encoder or physics simulator are directly used. This needs to be clarified in the main paper.
7. The paper mentions that the background masks are known when localising the objects via the encoder. If this is the case, the localisation problem becomes somewhat trivial. This should be clarified.
8. There needs to be a clear discussion on the limitations of the current approach — it does not scale to novel objects, needs to know the number of objects apriori and has not been shown to estimate object properties such as mass, friction etc.

Overall, the approach presents promising initial results towards an unsupervised method for modelling dynamical systems from video. There are several limitations that need to be made explicit and some additional experiments on more complicated systems and a few ablation studies can significantly improve the strength of the paper. I would suggest a borderline accept.

Typos:
1. Section 4.1, Setup: 5 values of (K, t_pred, t_ext) are given, need only 4
2. “K” is not introduced till the results section

Additional citation:
The following paper on learning physically consistent position and velocity representations for dynamical systems (and for use in visual MPC settings) should be cited:
Jonschkowski, Rico, et al. "Pves: Position-velocity encoders for unsupervised learning of structured state representations." arXiv preprint arXiv:1705.09805 (2017).

**Experience Assessment:**

I have published one or two papers in this area.

**Review Assessment: Checking Correctness Of Derivations And Theory:**

I assessed the sensibility of the derivations and theory.

**Review Assessment: Checking Correctness Of Experiments:**

I carefully checked the experiments.

**Review Assessment: Thoroughness In Paper Reading:**

I read the paper at least twice and used my best judgement in assessing the paper.

---

> ### Author Response · Authors · 2019-11-13
> **Addressing Reviewer 1's questions**
>
> Q1. A major limitation of the approach is the assumption that the equations governing the system are known. This makes it harder to generalise the system to novel tasks and more complicated settings with contact and object interactions. A potential way to overcome this could be to use ideas from prior work such as Interaction Networks where the dynamics are modelled as unary and binary interactions. While such dynamics models are black-box and not as interpretable compared to the current approach, they can easily generalise to novel tasks. Additionally, using positions and velocities as the latent state representation together with IN style transition models can be a sensible middle ground.
>
> Q1a: We would like to clarify this point, since it concerns the focal point of the paper.
> Our work is not meant to compete with black-box relational models in terms of model generality - quite the opposite. We tackle a very specific and well defined problem, which can be stated as: “Assuming we know the family of equations governing the system, but we only have videos, what model components are necessary in order to learn states, objects and physical parameters in an unsupervised way?” While other works learn objects and some learn states in an unsupervised way, *none* of them are able to answer the question, e.g, “In this gravitational system, what is the value of the gravity?” Our model is built specifically for this purpose, providing insight into the actual physical properties of the systems being analysed.
>
> Q1b. Additionally, in our IN baseline, explicit positions are part of the IN latent representation (since they are necessary for coordinate-consistent decoding). Thus we have already evaluated this middle ground and found it to perform poorly on our tasks.
>
>
> Q2. It would be great if there is an additional ablation experiment where the known equations of motion are replaced with a black-box neural network (while still retaining the position and velocity representation). This can quantify the effect of known dynamics and make the contributions of the paper (with regards to the decoder) more clear.
>
> Q2: The Interaction Network baseline used in Figure 2 and 3 serves precisely this purpose, since we keep the encoder and decoder used in the main model and simply replace the explicit physics with a relational black-box model. Thus PAIG vs IN+Inverse Graphics is exactly the requested ablation.
>
> Q3. An alternative way of generating consistent object positions from the encoder is to compute a mask-weighted average of the image co-ordinates. This can be a nice way of adding additional structure to the networks that can regularise training.
>
> Q3: We agree that it might be possible to generate coordinate consistent object positions in alternative ways. We used a spatial transformer since it provides an elegant way to combine shift, scale and rotation transformation.
>
> Q4. The approach uses a 3-layer MLP for generating velocity estimates — could this not be done via finite differencing? (e.g. higher-order backward differencing)
>
> Q4. Yes, it could. However, during experiments we noticed that using an MLP, though not an elegant solution, provided better initial velocity estimates, particularly in the gravitational system and pendulum cases. We believe this is because the initial velocity in this case depends on the acceleration (and higher moments) of the objects, which an MLP can easily capture, given object positions.
>
> Q5. Both the content and mask vectors are learnable but fixed for the entire task — i.e. it is not a function of the input image. This makes the approach not applicable to novel objects or even objects with minor color changes.
>
> A5. We address this in the main comment.
>
> (continues)

---

> ### Author Response · Authors · 2019-11-13
> **Addressing Reviewer 1's questions (cont.)**
>
> Q6. It is not clear how the translations, rotations and scale parameters for the Spatial Transformer are estimated. I presume that the positions and orientations predicted by either the encoder or physics simulator are directly used. This needs to be clarified in the main paper.
>
> A6: “the positions and orientations predicted by either the encoder or physics simulator” -> This is correct. They are passed to the spatial transformer according to Eq. (3), Pg 5.
>
> Q7. The paper mentions that the background masks are known when localising the objects via the encoder. If this is the case, the localisation problem becomes somewhat trivial. This should be clarified.
>
> A7. To clarify, the background masks are *not* known, they are learned templates like those of the remaining objects. We agree that using the phrasing “fixed background mask” in page 6 is misleading. We will change the phrasing to “learnable background mask”, to make it clearer.
>
> Q8. There needs to be a clear discussion on the limitations of the current approach — it does not scale to novel objects, needs to know the number of objects apriori and has not been shown to estimate object properties such as mass, friction etc.
>
> A8: We partly address this in the main comment.
> Regarding estimating additional object properties. The properties we chose to estimate, like gravitational and spring constants, are merely representative of what we can model, though this choice is arbitrary. For example, for the gravitational system, from Eq. (7) we can see that gravity and mass play the same role in controlling the force magnitude. That is, if we had set gravity as a known constant and mass as learned value, it would have worked just as well as the way we did it, which was to set mass as a known constant and gravity as a learned value.
>
> As mentioned previously, given that our objective is to perform system identification for a known system of equations, but given only video sequences, we do not see the need to know the number of objects apriori as a limitation. Our problem setting is not intended to compete with black-box dynamics models, but rather to show that it is possible to identify physical parameters such as gravity, spring constants, and more (although we demonstrate on a subset of examples, the proposed approach is applicable to general cases and parameters like friction or mass) from images, but that this requires multiple internal state representations be learned, including object representations and states. We have attempted to clarify this in the updated paper. The primary limitations of the proposed approach are in the rendering/ derendering components of the model, which still require significant improvement to scale to natural images and more complex scenes. We have included a limitations section to discuss these challenges in more detail.
>
> Q9: Typos and related work.
> A9: Thanks. We have corrected and added these accordingly.

---

### Official Review · AnonReviewer2 · 2019-10-24
**Official Blind Review #2**

**Rating:** 6

**Review:**

This paper presents a method for jointly making physical predictions and inferring latent physical parameters (e.g. gravity) in an unsupervised manner from video. Specifically, the proposed architecture consists of an object-centric encoder which estimates dynamic properties of each object in the scene (e.g. position), a differentiable physics engine, and a decoder.

I enjoyed reading this paper and think that it is a valuable contribution to the literature on physical reasoning, and thus lean towards acceptance. It elegantly combines several ideas that have been recently been investigated in the literature, though not yet put together. The experiments are well done and encompass not just prediction and inference but also control. The results look very impressive compared to existing models as well. However, my main critique would be that the evaluation domains are somewhat simplistic. If experiments in slightly more complex domains could be performed then I would be willing to increase my score to a full accept.

In the present paper, the scenes consist of only two or three objects, which is quite small compared to other papers in the literature which have evaluated on scenes with 6 or more objects. Similarly, given that the paper says it was inspired by the Physics 101 dataset, it is a bit disappointing that the proposed model was not evaluated on Physics 101 which would provide a more ecologically valid test of the model. At a minimum, I would like to see experiments with at least 6 objects if not more. Beyond that, including experiments on Physics 101 would change this from a good paper to an excellent paper.

I had a number of additional questions and comments:

I wondered how the proposed method would far at inferring object-specific latent parameters which cannot be inferred from images alone, such as friction or density. It seems like the velocity encoder could try to make these predictions as well, but it is not clear to me how well this would work with only a few frames.

Somewhat relatedly, it seems like a limitation of the method is that it would not work if the objects were not visually distinct (i.e. if the balls were all the same color)---in other words, it cannot track objects over time but must learn a fixed mapping of visual property (color) to slot. This is fine to leave for future work, but merits discussion. In particular, I suspect this is also related to why the IN results are so poor; due to small errors in the IN the objects end up out of the frame, and then because the model has no memory component, it just forgets about them. If the model had access to a memory that could track objects that leave the scene, then I expect the IN would fare much better. Similarly, I expect that even the model with the perfect simulator would fail in scenes in which objects can be fully occluded. I would appreciate if some discussion of these limitations could be added to the paper.

I am curious how well the model would perform if the number of object slots were not correct (i.e. if N is less then that actual number of objects, or more than the number of objects). It would be great if to include some experiments on this in the appendix.

The related work should probably mention the recent COBRA architecture [1], which also uses unsupervised scene decomposition combined with model-based RL.

Can you clarify whether the interaction net baseline is pretrained, or trained end-to-end with the encoder and decoder?

What are the errorbars over in Figure 5? Are they multiple seeds? If not, then I would like to see the figure updated with results from multiple training runs in order to properly assess variance.

Page 4: can you give more details on what a “fixed background mask” is?
Page 6: what is K? Is this supposed to be L (the number of frames used for velocity estimation?)
Page 6: when describing the values of (K, T_pred, T_ext), why are there 5 different settings?
The paper states earlier in the paragraph that there are only 4 different systems so I am a bit confused what these settings correspond to.

[1] Watters, N., Matthey, L., Bosnjak, M., Burgess, C. P., & Lerchner, A. (2019). COBRA: Data-Efficient Model-Based RL through Unsupervised Object Discovery and Curiosity-Driven Exploration. arXiv preprint arXiv:1905.09275.

**Experience Assessment:**

I have published in this field for several years.

**Review Assessment: Checking Correctness Of Derivations And Theory:**

I assessed the sensibility of the derivations and theory.

**Review Assessment: Checking Correctness Of Experiments:**

I carefully checked the experiments.

**Review Assessment: Thoroughness In Paper Reading:**

I read the paper thoroughly.

---

> ### Author Response · Authors · 2019-11-13
> **Addressing Reviewer 2's questions**
>
> Q1: I wondered how the proposed method would far at inferring object-specific latent parameters which cannot be inferred from images alone, such as friction or density. It seems like the velocity encoder could try to make these predictions as well, but it is not clear to me how well this would work with only a few frames.
>
> A1: Please note that the proposed approach is able to learn object-specific latent parameters like mass, gravity, or spring constants that are not immediately detectable from images alone, as it relies on the structural supervisory bias of the known physical equations to learn about parameters of interest from video. This is typically a data intensive process, however, as it requires that object representations/masks be learned in addition to parameter estimation and at the same time as object state estimation, and the proposed approach is unable to learn these properties from merely a few images. Moreover, we assume that the video sequence is exciting enough to allow for these parameters to be learned - eg, we cannot hope to learn about gravity from a stationary object, but can say something about this if we see a pendulum spinning, and know something about the equations of motion of a pendulum.
>
> Q2: Somewhat relatedly, it seems like a limitation of the method is that it would not work if the objects were not visually distinct (i.e. if the balls were all the same color)---in other words, it cannot track objects over time but must learn a fixed mapping of visual property (color) to slot. This is fine to leave for future work, but merits discussion. In particular, I suspect this is also related to why the IN results are so poor; due to small errors in the IN the objects end up out of the frame, and then because the model has no memory component, it just forgets about them. If the model had access to a memory that could track objects that leave the scene, then I expect the IN would fare much better. Similarly, I expect that even the model with the perfect simulator would fail in scenes in which objects can be fully occluded. I would appreciate if some discussion of these limitations could be added to the paper.
>
> A2: This is a good point, and we will add some discussion regarding the need for visually distinct objects and limitations due to occlusions. However, it should be noted that it is unlikely that  IN fail because of lack of memory, since DDPAE does have memory (the recurrent component is an LSTM) and the long term dynamics are still incorrect. We believe both of these models fail because if a certain dynamic regime (say, gravitational whiplash during close contact between the objects) was not seen during training, these black-box motion predictor methods simply do not have the ability to generalize correctly, whereas using the underlying physics equations provides the necessary inductive bias to do so.
>
> Q3: I am curious how well the model would perform if the number of object slots were not correct (i.e. if N is less then that actual number of objects, or more than the number of objects). It would be great if to include some experiments on this in the appendix.
>
> A3: We ran this experiment for the gravitational system and included it in appendix Sec E in the revised paper. In short, if N > N_true, it detects N_true objects, and leaves an empty mask slot, if N < N_true, it detects N objects. But the frame prediction accuracy of both models is reduced because the internal physical simuation is now inaccurate.
>
> (continues)

---

> > ### Comment · AnonReviewer2 · 2019-11-14
> > **Response to authors**
> >
> > Thanks for your response and for the clarifications.
> >
> > Were you able to run any experiments with larger numbers of objects than 2 or 3? As I mentioned in my review, I did not think it was essential to test on Physics 101; however, I do think it is important to test on similar numbers of objects as has been done in past work.
> >
> > 1. The way mass, gravity, and spring constants are defined in the paper are not really "object-specific"---they are the same value for all objects and thus are object-agnostic. I was curious about values that would be different for each object (e.g., one object has a mass of 1 and another has a mass of 2), and what changes would be required to allow such values to be inferred.
> >
> > 2. Thanks for adding some discussion about this. That's a good point about DDPAE having memory, though I still wonder if the IN would fare much better with memory since its structure more accurately matches the computation of a physics engine than DDPAE does (so I would expect the IN to be able to better model the dynamics than DDPAE in general---which seems to indeed be true from your results).
> >
> > 3. Thanks for including these results! It's good that it seems to at least be somewhat robust to misspecification of the number of slots, though it's a bit unfortunate that if there are too many slots it cannot ignore the missing slots. As a topic for future work, perhaps some sort of attention mechanism could help here.

---

> > > ### Author Response · Authors · 2019-11-15
> > > **Response**
> > >
> > > Thank you for taking the time to review our responses.
> > >
> > > We decided against running these experiments since we believe that most systems of interest where we are looking to estimate physical parameters have at most 3 objects (vide Physics101). While we could have simply made a 5-object bouncing ball dataset, we believe this would be of little interest to validate the main point of the paper.
> > >
> > > 1. You are correct. We have used equal values for every object, but we believe that using a different learnable parameter for each object would work without further modifications.
> > >
> > > 2. It is entirely possible that IN with memory would fare better. Perhaps a recurrent structure akin to that used in Visual Interaction Networks would improve results, though we believe the extrapolation results are representative of any black-box model.
> > >
> > > 3. There are many possibilities for future work in this direction. An attention mechanism is one. Another one could be doing some sort of likelihood-based model selection (though this requires training several models in parallel) to pick how many objects are in the scene.

---

> ### Author Response · Authors · 2019-11-13
> **Addressing Reviewer 2' questions (cont.)**
>
> Q4: The related work should probably mention the recent COBRA architecture [1], which also uses unsupervised scene decomposition combined with model-based RL.
>
> A4: Thank you for pointing to that reference, we will include it in the related work section.
>
> Q5: Can you clarify whether the interaction net baseline is pretrained, or trained end-to-end with the encoder and decoder?
>
> A5: The interaction net baseline is trained end-to-end with the encoder and decoder in order to obtain a fair and direct comparison with the physics model.
>
> Q6: What are the errorbars over in Figure 5? Are they multiple seeds? If not, then I would like to see the figure updated with results from multiple training runs in order to properly assess variance.
>
> Q6: Yes. The error bars are the 95% confidence interval across 50 test runs with random initialization seeds. We use this setting in order to make our performance directly comparable to that of PlaNet, in which the authors use this evaluation metric (though with fewer runs).
>
> Q7: Page 4: can you give more details on what a “fixed background mask” is?
>
> A7: A fixed background mask is a learnable mask which, unlike the remaining object masks, is not subject to affine transformation (it is input independent). We agree the term is misleading and will rephrase this as “learnable background mask”.
>
> Q8: Page 6: what is K? Is this supposed to be L (the number of frames used for velocity estimation?)
>
> A8: Yes, it is supposed to be L. We will correct this typo.
>
> Q9: Page 6: when describing the values of (K, T_pred, T_ext), why are there 5 different settings?
> The paper states earlier in the paragraph that there are only 4 different systems so I am a bit confused what these settings correspond to.
>
> A9: The 5th setting here corresponds to the 2 MNSIT digits connected by a spring, on a CIFAR background. We consider 2-balls spring and 2-digit spring as the same physical system, so we count only 4 different systems from a dynamics point of view.

---

### Author Response · Authors · 2019-11-13
**Main comment to all reviewers**

Thank you for taking the time to review our work, we are grateful for the comments and believe that addressing these have significantly improved our paper. We reply to each reviewers questions in the respective thread, and answer the simplicity concern expressed by all reviewers here.

We agree with the general sentiment the reviewers express with regards to the simplicity of the settings used. In this work we focused on putting together the building blocks that would allow us to integrate physics engines with encoder-decoder architectures for learning physical parameters from pixels. As we found during our research and experimentation, this posed sufficient difficulties that even developing a model that works on these datasets proved challenging. Even though Physics101 is a natural real video application of our model, we would have to add extra components to the model in order to deal with varying object appearance, namely a tracker (for persistent object identity), a visual latent encoding (in order to allow decoding), and a dynamic background model (for moving humans in the background). While these are exciting challenges to solve, and we plan to pursue this research direction, they would make the model increasingly complex and would detract from clearly describing the key contributions that make physics-as-inverse-graphics possible: coordinate-consistent decoders, autoencoder losses and differentiable physics engines. We have added a Limitations section to the reviewed paper in order to make the model shortcomings clear and provide avenues for future work.

---

### Decision · Program_Chairs · 2019-12-19

**Decision:**

Accept (Poster)

**Comment:**

The submission presents an approach to estimating physical parameters from video. The approach is sensible and is presented fairly well. The main criticism is that the approach is only demonstrated in simplistic "toy" settings. Nevertheless, the reviewers recommend (weakly) accepting the paper and the AC concurs.